# Application of Procalcitonin for the Rapid Diagnosis of *Clostridioides difficile* Infection in Patients with Inflammatory Bowel Disease

**DOI:** 10.3390/diagnostics12123108

**Published:** 2022-12-09

**Authors:** Shuhua Xie, Peisong Chen, Dong Wang, Xiaobing Jiang, Zhongwen Wu, Kang Liao, Min Liu, Shihong Zhang, Yili Chen

**Affiliations:** Department of Laboratory Medicine, The First Affiliated Hospital, Sun Yat-sen University, Guangzhou 510080, China

**Keywords:** *Clostridioides difficile*, inflammatory bowel disease, PCT, CRP, WBC

## Abstract

**Background**: The incidence of *Clostridioides difficile* infection (CDI) has increased in recent years in patients with inflammatory bowel disease (IBD). *C. difficile* is a toxin-producing bacterium, and CDI results in the worsening of underlying IBD, increasing the risk of IBD treatment failure, surgery, and hospitalization. Because the symptoms of CDI overlap with those of IBD, it is challenging to make a differential diagnosis. Therefore, early, rapid, and reliable diagnostic tools that can identify CDI in IBD patients would be valuable to clinicians. **Methods**: This study retrospectively collected 135 patients with IBD. Among them, 44 patients were diagnosed with CDI, and 42 patients were diagnosed with viral or fungal infections. A total of 49 patients without infections were defined as the control group. The diagnostic values of procalcitonin (PCT), C-reactive protein (CRP), and white blood cell (WBC) count in the peripheral blood were examined. **Results**: In this study, PCT levels were significantly higher in patients with CDI than in non-CDI patients (including patients with viral/fungal infections and the control group; *p* < 0.001 and *p* < 0.05, respectively). CRP levels were significantly higher in patients with CDI than in non-CDI patients (*p* < 0.05). The area under the curve (AUC) of PCT and WBC count were compared using DeLong’s test: the AUCs of PCT vs. CRP for the detection of the IBD–CDI group and the control group was 0.826 [95% confidence interval (CI) 0.743–0.909] vs. 0.663 [95% confidence interval (CI) 0.551–0.774] (*p* < 0.05), respectively. WBC count was inferior as a diagnostic tool for CDI. The sensitivity was 59.09% (95% CI: 43.2% to 73.7%), the specificity was 89.80% (95% CI: 77.8% to 96.6%), and the positive likelihood ratio LR (+) was 5.79 for PCT for the diagnosis of CDI. **Conclusions**: The present study demonstrates the superiority of PCT over CRP and WBC count for the rapid diagnosis of CDI in IBD patients.

## 1. Background

Inflammatory bowel disease (IBD), including Crohn’s disease (CD) and ulcerative colitis (UC), is a group of chronic disorders of the gastrointestinal tract that is characterized by inflammation of an unknown etiology, and the recurrence rate is high. The clinical characteristics are diverse and include serious complications, such as bleeding, perforation, and abscess formation [1,2].

The resident microorganisms in the human gastrointestinal tract play an important role in maintaining intestinal equilibrium. There are several mechanisms, including the degradation of xenobiotic substances, synthesis of beneficial metabolites and vitamins, immune system regulation, and colonization resistance against invading pathogenic microorganisms [3]. Chronic diarrhea caused by IBD leads to dysbacteriosis, decreasing bacterial diversity in the long term and reducing colonization resistance against *Clostridioides difficile* [4]. Changes in the community structure of the gut microbiota and markedly decreased microbial diversity have been identified in patients with CDI [5]. The long-term use of immunosuppressive medications for the treatment of IBD, such as immunomodulators and glucocorticoids, particularly when these agents are used in combination, result in an increased risk of opportunistic infections. Individuals with IBD have a nearly five-fold-higher risk of *C. difficile* infection than individuals without IBD [6]. In contrast, infections with bacteria other than *C. difficile* are uncommon in IBD patients experiencing a disease flare [7].

*C. difficile*, a Gram-positive spore-forming anaerobe, produces exotoxins that cause a spectrum of diseases. Serious infections are occasionally complicated by toxic megacolon, perforation, sepsis, and death [8]. It is reported that CDI complicates the course of IBD, leading to longer hospital stays, an increased rate of colectomy, and increased mortality [9]. CDI and IBD have an obvious overlap with the clinical manifestations of colitis. Diarrhea is the most prominent symptom, but bleeding is more likely in IBD; other common symptoms include abdominal discomfort and fever. Therefore, it is difficult to clinically distinguish an acute IBD attack and acute CDI. In all IBD patients, these symptoms or signs may indicate a colitis attack, such as increased blood in the stool. Accordingly, these patients should also be tested for the presence of toxigenic *C. difficile* in the stool. This presents challenges in diagnosis and treatment.

Based on the above evidence, the 2013 ACG guidelines made the following conditional recommendation: in IBD patients with severe colitis, the simultaneous initiation of empirical therapy directed against both CDI and IBD attacks may be needed while awaiting the results of *C. difficile* testing. The escalation of immunosuppression medications should be avoided if there is a possibility of untreated CDI [10]. Therefore, a fast and reliable CDI diagnostic tool is valuable for clinicians.

Nucleic acid amplification tests (NAATs) have a sensitivity greater than 95% for the diagnosis of CDI [11,12]. The positive predictive value is dependent on the presence of diarrhea [13,14]. *C. difficile* can colonize without producing significant amounts of toxin, resulting in a symptomatic infection and making a positive PCR result clinically irrelevant [15,16]. An ideal test to diagnose CDI in a situation in which symptoms are not reliable would be a highly sensitive assay to measure the level of the toxin [17]. Unfortunately, most enzyme immunoassays (EIAs) for toxins have low sensitivities, sometimes even in the 50% to 70% range. Glutamate dehydrogenase (GDH) is produced by all *C. difficile* strains, regardless of whether they are toxigenic, and it cannot be used to identify toxigenic strains of *C. difficile* [18]. Currently, using PCR/NAATs alone leads to overdiagnoses, and the use of EIAs for toxins alone may lead to underdiagnoses [19]. Therefore, reliable, specific, and sensitive markers are needed for CDI in IBD patients.

At present, routine inflammation markers, including procalcitonin levels, C-reactive protein levels, and white blood cell count, are widely used for the diagnosis of infectious diseases. C-reactive protein (CRP) is one of the most important proteins involved in acute inflammation. CRP has a short half-life, so CRP levels increase and decrease rapidly during acute inflammation [20]. The detection of CRP is inexpensive and rapid. Procalcitonin (PCT) is a prohormone of calcitonin, and serum levels of PCT are mainly increased in patients with bacterial infections. PCT has been found to reach a high level within 8–24 h after the onset of a bacterial infection [21]. As an inflammatory marker, PCT has obvious advantages because it is stable, not affected by exogenous bacteria in vitro, and the test is relatively easy to carry out with a fast turnaround time of 2 h [5].

This is a retrospective study. We aimed to determine the efficacy of noninvasive and rapid routine inflammation marker tests in monitoring IBD patients infected with *C. difficile.* Our research aimed to provide auxiliary evidence of CDI infection in IBD patients and carry out antibiotic treatment as early as possible to improve prognosis. Thus, in the present study, we evaluated the diagnostic value of PCT, CRP, and WBC count for CDI in IBD patients.

## 2. Methods

### 2.1. Patient Eligibility and Classification

In this study, a total of 445 patients diagnosed with inflammatory bowel disease (IBD) were collected from the medical records of the First Affiliated Hospital of Sun Yat-sen University register from 2016 to 2020. *C. difficile* should be detected in IBD patients with a high risk of CDI, including the following: (1) all active IBD inpatients; (2) IBD patients with diarrhea in remission or recent exposure to risk factors (such as contact with CDI patients, gastrointestinal surgery, tube feeding, bowel preparation, etc.); (3) IBD patients with severe colitis, without bacteriological evidence, and requiring empirical CDI treatment; and (4) patients with low immunity, diabetes, renal failure, malnutrition, etc. An endoscopy was performed in all individuals. IBD lacks the gold standard for diagnosis and needs to be comprehensively analyzed in combination with clinical manifestations, laboratory examinations, endoscopies, imaging examinations, and histopathological examinations. The Crohn’s disease activity index (CDAI) is used to evaluate the severity of disease activity. The improved ‘Truelove’ and ‘Witts’ disease severity classification criteria were used to grade the active stage of UC [22]. Patients diagnosed with inflammatory bowel disease (IBD) participated in the study, and 135 patients were included. A total of 300 subjects were excluded from the analysis due to incomplete blood tests (n = 270), the presence of other positive bacteria cultures in the stool (n = 10), and other sites of bacterial infection (n = 20). Among all of the included 135 patients, 44 patients were diagnosed with CDI based on NAATs for *Clostridioides difficile*. Forty-two patients were diagnosed with other opportunistic infections, including viral infections and fungal infections. Viral infection diagnosis was based on positive viral IgM antibodies and positive viral PCR results in blood samples, including Epstein–Barr virus (EBV), cytomegalovirus (CMV), herpes simplex virus type 1 (HSV-1), and herpes simplex virus type 2 (HSV-2); meanwhile, fungal infection diagnosis was based on positive stool culture results, mostly of Candida infection. Forty-nine IBD patients without infections were enrolled in the control group (Figure 1). All methods were carried out in accordance with the declaration of Helsinki. Written informed consent was obtained from all subjects. This study was approved by the Clinical Research and Ethics Committee of the First Affiliated Hospital of Sun Yat-sen University.

A six-component ATLAS score, which is useful in stratifying the severity of CDI patients [23], was measured on the day of entry into the study (within 48 h of a positive NAAT for *Clostridioides difficile*). The components of ATLAS score were as follows: age (in years); treatment with systemic antibiotics (which occurred on one or more days of CDI therapy); temperature (in degrees Celsius); total leukocyte count; serum albumin; and serum creatinine as a measure of renal function.

### 2.2. Detection of Clostridioides Difficile

CDI was diagnosed based on positive nucleic acid amplification tests (NAATs) for the *Clostridioides difficile* tcdA, tcdB, cdtA, and cdtB genes (GeneXpert C. difficile Assay, Cepheid, Sunnyvale, CA, USA).

### 2.3. Laboratory Measurement of Inflammatory Markers

Serum PCT levels were determined by a quantitative electrochemiluminescence immunometric assay using a COBAS E601 analyzer (e601 module of the Cobas 6000 system, Roche, Switzerland) with a level of <0.05 ng/mL considered normal. Plasma CRP levels were measured by immunoturbidimetry using a Mindray CRP-M100 analyzer (Mindray CRP-M100, Shanghai, China) a Dade Behring BN II (Marburg, Germany). The reference value provided by the manufacturer was < 10 mg/L. Whole WBC levels were measured by a Mindray BC-6800 analyzer (Mindray BC-6800, Shenzhen, China) and SysmexXN-9000 analyzer (SysmexXN-9000, Kobe, Japan) with a level of <10 × 10^9^/L considered normal.

### 2.4. Statistical Analysis

Statistical analyses and graphic presentations were carried out using the software program GraphPad Prism (Version 5.0, GraphPad Software Inc., San Diego, CA, USA), MedCalc (Version 20.1116, Ostend, Belgium) and IBM SPSS software (Version 25.0, SPSS China). Patient characteristics among the groups were analyzed using Pearson’s chi-square test, one-way ANOVA, and Tukey’s multiple comparison test. Categorical variables were expressed in proportions. Infection marker variables were expressed as the mean ± SEM. DeLong’s test was used for the pairwise comparison of AUC and ROC curves. AUCs (with 95% confidence intervals) were calculated to assess the diagnostic values of the tests; AUCs > 0.70 were considered clinically relevant. Youden’s index, sensitivity, specificity, and the LR (+) were calculated with MedCalc. In all analyses, statistical tests were two-sided, and *p* values of less than 0.05 were considered statistically significant.

## 3. Results

### 3.1. Patient Characteristics

Of these 135 IBD patients, 83 patients were diagnosed with Crohn’s disease and 52 patients were diagnosed with ulcerative colitis. Age and *Clostridioides difficile* infection incidences were different in CD and UC patients (*p* < 0.001 and *p* < 0.05, respectively), while sex and colectomy rate were similar in CD and UC patients (*p* > 0.05). Crohn’s disease patients had a younger median age but a higher CDI incidence than ulcerative colitis patients. The clinical characteristics of CD and UC patients are shown in Table 1.

### 3.2. Evaluation of Diagnostic Value

According to the clinical infection of *Clostridioides difficile*, virus and fungus, the IBD patients were divided into three groups. The sex ratio, clinical characteristics, and outcomes are shown in Table 2. Sex was different in the distribution of the three groups (*p* < 0.005). There was no difference in endoscopic activity and colectomy among the three groups (*p* > 0.05).

The level of inflammatory biomarkers in the different groups in the study are shown in Table 3 and Table 4 and Figure 2. The mean of PCT was significantly higher in IBD–CDI patients than in non-CDI patients, as well as in IBD patients with viral or fungal infections (both *p* < 0.001). The mean of CRP was significantly higher in IBD–CDI patients than in IBD patients (*p* < 0.05), and the mean of WBC count was not significantly different among the three groups (*p* > 0.05).

The receiver operating characteristic (ROC) curves corresponding to each infection biomarker in the IBD–CDI group and the IBD control group for the prediction of CDI in IBD patients are shown in Figure 3. Based on the AUC of the ROC curve, PCT had an AUC of 0.826 (95% CI: 0.743–0.909, *p* = 0.000). CRP had an AUC of 0.663 (95% CI: 0.551–0.774, *p* = 0.007). PCT was more accurate in the diagnosis of CDI or being noninfectious than CRP (DeLong’s test, *p* = 0.0106).

The receiver operating characteristic (ROC) curves corresponding to each infection biomarker in the IBD–CDI group and the viral/fungus group for the prediction of CDI infections in IBD patients are shown in Figure 4. Based on the AUC of the ROC curve, PCT had an AUC of 0.716 (95% CI: 0.609–0.824, *p* = 0.001). CRP had an AUC of 0.615 (95% CI: 0.496–0.735, *p* = 0.067). PCT was more accurate in the diagnosis of CDI or viral/fungus infections than CRP (DeLong’s test, *p* = 0.170).

Youden’s index, sensitivity, specificity, and the LR (+) were calculated with MedCalc software. Youden’s index associated criterion value >0.09, the sensitivity%, specificity%, and positive likelihood ratio (LR+) predictive values of the best PCT cutoff value for the diagnosis of *Clostridioides difficile* infection are shown in Table 5. The sensitivity% was 59.09% (95% CI: 43.2% to 73.7%), the specificity% was 89.80% (95% CI: 77.8% to 96.6%), and the LR (+) was 5.79.

The ATLAS scoring system was used to stratify the severity of CDI patients in this study, which divided CDI patients into seven groups (scores of 0 to 6). PCT and the rate of colectomy were calculated for each stratification, which are shown in Table 6.

## 4. Discussion

IBD patients are at increased risk of developing CDI with worse clinical consequences than the rest of the population. European articles have reported that IBD patients with CDI have an elevated 1.2- to 3-fold risk of gastrointestinal surgery or emergent colectomy [22]. IBD patients are at high risk of CDI recurrence [23]. Crohn’s disease patients who were younger at diagnosis are much more likely to obtain CDI than ulcerative colitis patients. The striking part of the epidemiology of CDI is an increase in individuals who are younger [6,24].

A positive stool test for *C. difficile* or its toxins does not absolutely diagnose CDI. The sensitivity of stool culture is limited; it is time-consuming and does not distinguish infection from colonization. NAAT detection methods are more sensitive and can quickly obtain results. NAATs are considered superior to other methods, but it is difficult to accurately distinguish the colonization and infection of *C. difficile*, which may lead to excessive diagnoses and treatments of CDI. The two-step method, namely GDH and toxin A/B, is recommended in the bacterial-infection-diagnosis guidance document [25]. In the three-step method, GDH EIAs or NAATs are used for the initial screening, and A/B EIA tests are used when the initial screening is positive. If the A/B EIAs test is negative, cultures or NAATs are used to confirm the identification [26]. However, these tests are time-consuming, multi-step, and costly.

C-reactive protein is one of the most important proteins in acute inflammation. It is maintained at a low level (less than 1 mg/L) in peripheral blood secreted by hepatocytes in healthy people, but it increases when acute inflammation occurs and is induced by interleukin-6 (IL-6), tumor necrosis factor-*α* (TNF-*α*), and interleukin-1*β* (IL-1*β*). The CRP value sharply increases, even reaching up to three- to four-hundred times its usual value. CRP remains at a level of 10–40 mg/L, indicating virus infection or chronic inflammation [10]. In this study, we determined that IBD patients suffered from chronic inflammation with a mean CRP level of nearly 37 mg/L and IBD patients with viral or fungal CRP levels remained at a mean of nearly 40 mg/L, while IBD patients suffered from *C. difficile* infection. The mean CRP level increased up to nearly 60 mg/L, nearly 1.5 times higher than that of non-CDI patients.

Procalcitonin is a prehormone of calcitonin. When bacterial infections occur, the circulation levels of PCT increase, mainly due to the presence of bacterial endotoxins and exotoxins, as well as inflammatory cytokines, such as TNF, IL-2, and IL-6. When viral infections occur, interferon-gamma (IFN-γ) is released. PCT production is inhibited by IFN-γ [27]. High levels are found in severe bacterial infections and low levels are found in nonspecific inflammatory diseases and viral infections; therefore, PCT has been considered a promising marker for the diagnosis of bacterial infections [28]. PCT may be used to support clinical decisions in different types of infections, marking the beginning or end of antibiotic therapy [29].

In our study on IBD patients, the purpose of monitoring PCT was to stop immunosuppressive drug treatment as soon as possible and relatively reduce the duration of immunosuppressive treatment. In this present study, we found that mean PCT levels in serum increased nearly three-fold in IBD–CDI patients compared to non-CDI patients with IBD, considering PCT requires more intense stimuli to increase compared with CRP [30]. PCT is characterized by a relatively high specificity but low sensitivity in abdominal local infection diseases, which differentiates bacterial infection from other systemic inflammation or viral infections [28]. In this study, PCT showed a low sensitivity but high specificity for CDI diagnosis in IBD patients as well. The PCT values are relatively high in moderate and severe CDI, suggesting that PCT could be more useful in severe CDI diagnoses, and that more careful differential diagnoses are needed in mild cases. However, more cases need to be collected in the future to clarify the correlation between PCT and CDI severity. Briefly, PCT provides a reference for the clinical withdrawal of immunosuppressants and the initiation of antibiotic therapy in IBD patients. As a marker of inflammation, PCT is stable, easy to operate, specific, and fast.

We found that CRP and WBC count were not ideal diagnostic tools for identifying *C. difficile* infections or other viral and fungal infections in IBD patients. Although CRP levels increased in the CDI of IBD patients, based on the pairwise comparison of AUC and ROC curves by DeLong’s test, PCT was more accurate in the diagnosis of CDI than CRP.

Several limitations should be considered in the present study. First, given that it was a retrospective study with a small number of patients, the effect of confounding variables cannot be ruled out. Second, we did not have a long-term follow-up of mortality and recurrence rate in patients with CDI. Third, there were only a few cases detected by EIA in these retrospective studies, so the EIA results were not included in the study.

## 5. Conclusions

Overall, the present study demonstrates the superiority of PCT over CRP and WBC count in differentiating CDI in IBD patients. PCT can be used as a biomarker for helping with the rapid diagnosis of CDI in IBD patients. Whether PCT levels can help predict the clinical outcome and progress of IBD patients’ needs further research to fully evaluate.

## Figures and Tables

**Figure 1 diagnostics-12-03108-f001:**
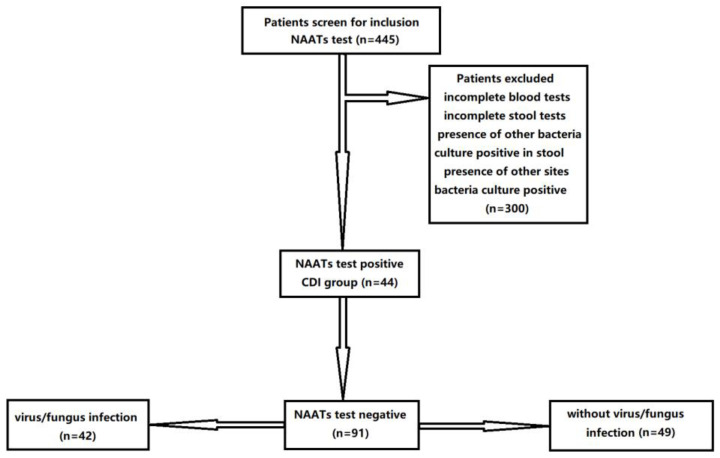
Study algorithm. IBD: inflammatory bowel disease; CDI: Clostridioides difficile infection.

**Figure 2 diagnostics-12-03108-f002:**
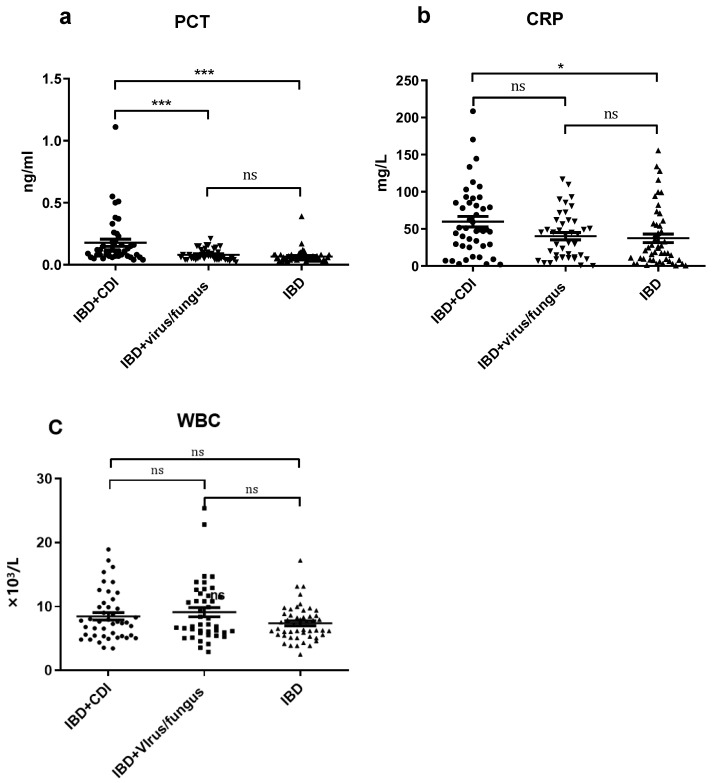
Scatter plots of PCT (**a**), CRP (**b**), and WBC count (**c**) for patients with inflammatory bowel disease with or without *Clostridioides difficile* infection or with viral/fungal infections. (**a**,**b**) In cases of PCT, the mean was obviously higher in IBD–CDI patients than in patients without CDI or with viral/fungal infections. In cases of CRP, the mean was obviously higher in IBD–CDI patients than in patients without CDI infections. (**c**) There was no significant difference in WBC count among patients with or without *Clostridioides difficile* infection or with viral/fungal infections. * *p* < 0.05, *** *p* < 0.001, ns: no significance.

**Figure 3 diagnostics-12-03108-f003:**
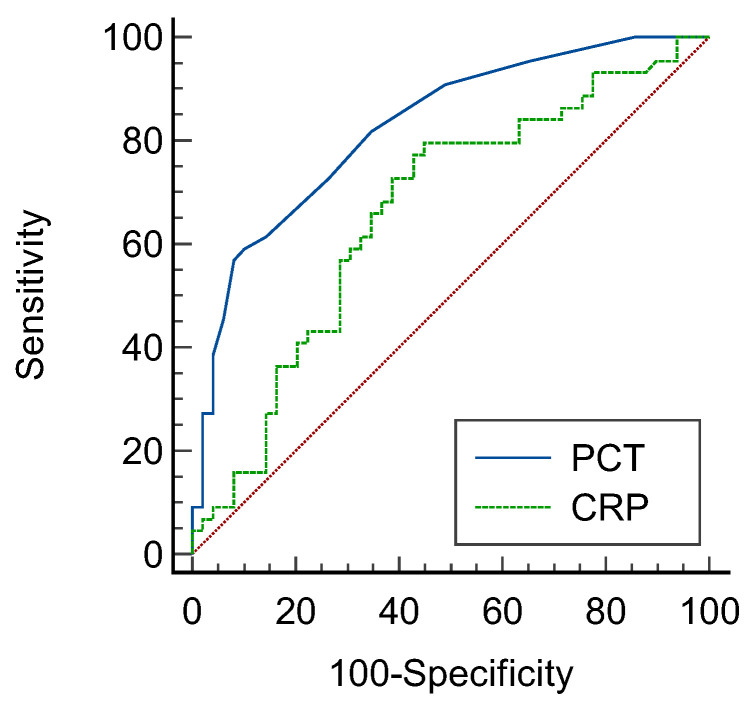
Prediction of *Clostridioides difficile* infections by ROC curve analysis (CDI vs. IBD control group). PCT with an AUC of 0.826 (95% CI: 0.743–0.909, *p* = 0.000). CRP with an AUC of 0.663 (95% CI: 0.551–0.774, *p* = 0.007). DeLong’s test, *p* = 0.0106.

**Figure 4 diagnostics-12-03108-f004:**
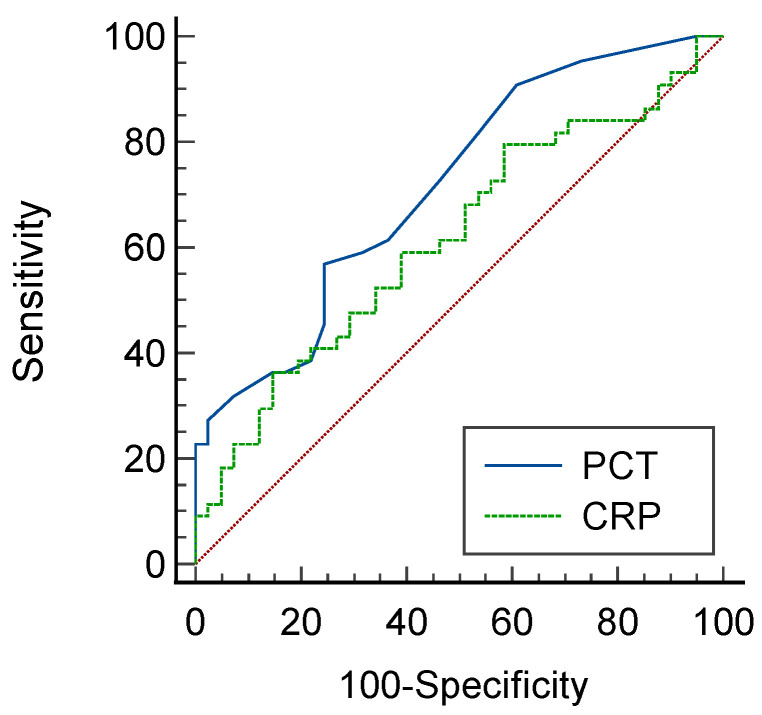
Prediction of *Clostridioides difficile* infections by ROC curve analysis (CDI vs. viral/fungus group). PCT with an AUC of 0.716 (95% CI: 0.609–0.824, *p* = 0.001). CRP with an AUC of 0.615 (95% CI: 0.496–0.735, *p* = 0.067). DeLong’s test, *p* = 0.170.

**Table 1 diagnostics-12-03108-t001:** Clinical characteristics of CD and UC patients.

Characteristic	Crohn’s Disease (n = 83)	Ulcerative Colitis (n = 52)	*p*-Value
Age, y, median (IQR)	27.0 (15–61)	49.5 (18–70)	<0.0001 ***
Male, n (%)	50 (60.2)	28 (53.8)	0.391
CDI incidence, n (%)	32 (38.6)	12 (23.1)	0.014 *
Colectomy, n (%)	18 (21.7)	6 (11.5)	0.056

* *p* < 0.05; *** *p* < 0.001.

**Table 2 diagnostics-12-03108-t002:** IBD patients’ characteristics in different infection groups.

Chi-Square Test	Male, n (%) *p*-Value	Colectomy, n (%) *p*-Value	Active Period, n (%) *p*-Value
IBD+CDI vs. IBD+ virus/fungus	0.031 *	0.675	0.675
IBD+CDI vs. IBD	0.127	0.181	0.520
IBD+ virus/fungus vs. IBD	0.476	0.369	0.290

* *p* < 0.05.

**Table 3 diagnostics-12-03108-t003:** Patients’ inflammatory biomarker levels of IBD patients in different groups.

Characteristic	IBD with CDI (n = 44)	IBD with Viral/Fungus Infections (n = 42)	IBD Control Group (n = 49)	One-Way ANOVA *p*-Value
PCT (ng/mL) Mean ± SEM	0.178 ± 0.029	0.081 ± 0.007	0.066 ± 0.007	<0.0001 ***
CRP (mg/L) Mean ± SEM	59.70 ± 7.01	40.14 ± 4.91	37.35 ± 5.71	0.0177 *
WBC count (×10^9^/L) Mean ± SEM	8.48 ± 0.57	8.93 ± 0.71	7.38 ± 0.39	0.130

* *p* < 0.05; *** *p* < 0.001.

**Table 4 diagnostics-12-03108-t004:** Comparison of inflammatory biomarkers between groups of IBD patients.

Tukey’s Multiple Comparison Test	PCT *p*-Value	CRP *p*-Value	WBC *p*-Value
IBD+CDI vs. IBD+ virus/fungus	0.821	0.066	0.840
IBD+CDI vs. IBD	<0.001 *	0.022 *	0.345
IBD+ virus/fungus vs. IBD	<0.001 *	0.942	0.126

* *p* < 0.05.

**Table 5 diagnostics-12-03108-t005:** Diagnostic performance indices of PCT in *Clostridioides difficile* infection of IBD.

Sensitivity%	95%CI	Specificity%	95%CI	LR+
59.09%	43.2–73.7%	89.80%	77.8–96.6%	5.79

**Table 6 diagnostics-12-03108-t006:** ATLAS score, PCT, and rate of colectomy of *Clostridioides difficile* infection patients.

ATLAS Score	Number of Patients	PCT	Rate of Colectomy
		Mean ± SEM (ng/mL)	n (%)
0	11	0.1255 + 0.0285	2 (16.7%)
1	17	0.1888 + 0.0630	3 (17.6%)
2	6	0.1783 + 0.0715	1 (16.7%)
3	6	0.2317 + 0.0766	2 (33.3%)
4	2	0.1000 + 0.0600	1 (50%)
5	1	0.3800	0
6	1	0.1800	1 (100%)

## Data Availability

All data generated or analyzed during this study are included in this published article.

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
