# Peer review of "Application of Procalcitonin for the Rapid Diagnosis of Clostridioides difficile Infection in Patients with Inflammatory Bowel Disease"

_diagnostics, 2022, doi:10.3390/diagnostics12123108_

Round 1

Reviewer 1 Report

Xie et al present a study demonstrating the specific nature of serum PCT for the diagnosis of CDI in IBD. This is an unmet need as they state in the introduction that clinically differentiating CDI from patients with IBD is difficult. They show the value of serum PCT which is clinically appropriate. The selection of patient needs to be better expanded as currently I do not see that this study looks at all comer patients testing for CDI.

Major revisions:

Major: Methods: The first paragraph of the methods reports that patients were approached for this study. This would suggest that it was a prospective study where blood and stool was collected. However at the beginning it discusses critera for testing of C. difficile. Please specify if this is a prospective or retrospective study. From what I see this is retrospective including patients that were clinically suspected and therefore tested for C. difficile.

Major: Methods: What is the value of culture if the NAAT was performed. If NAAT is highly sensitive, culture will not add anything to this study. Simply limit the study to NAAT positive. Was cytotoxin testing performed with culture? This would potentially add value.

Major methods: How many CDI tests were performed in total. If 44 were positive and assuming 10% of patients with IBD have CDI, then > 300 patients should have an available negative test. I understand that some might not have relevant blood testing and were included, but for Figure 1, please start with how many CDI tests were performed. From there how many don't have relevant bloods data. From there how many are positive and negative. Among negative how many have other viral and fungal infections.

Major: Methods: What are the clinical criteria assessed? What does active period mean? What time frame was the colectomy performed in. How was endoscopic activity assessed? Was endoscopy performed in all individuals?

Major: Results: For Table 2, is there a difference in IBD type across the groups.

Major: For Table 2, my understanding is that one-way ANOVA was performed. But were any comparisons performed for specific groups (i.e: between CDI and viral, between CDI and control, between viral and control? Lower p-value threshold would be required here). These comparisons are shown in Figure 2, but how they were measured should be reported in the methods better.

Major: The discussion states that Crohns patients were more likely to undergo colectomy compared to UC. This intuitively is not what one would expect and frankly this is also not reported in the results.

Major: A limitation should be added. The authors frequently discuss differences in testing technique, but unfortunately the study only used NAAT which is sensitive but not specific. One would image if direct testing against toxin were performed that differences in CRP and PCT would be more pronounced and therefore more valuable.

Minor:

Line 3 of the introduction: States recurrence rate is high, but does not define recurrence of what? GI symptoms, flares or CDI?

Minor: Results – if the AUC curves are being compared, can this be tested for significance (Delongs test).

Reviewer 2 Report

Clostridium difficile should be Clostridioides difficile.

Biomarkers for inflammatory response may vary depending on the severity of CDI. Because C. difficile carriage is common even in healthy individuals, C. difficile positivity does not equate to CDI, and the diagnosis is often difficult to make in clinical practice.

The procalcitonin values in this study are relatively high and less sensitive.

Will this study include a lot of severe CDI?

Please consider the severity of CDI, including Alb, Cr, ATLAS score, etc.

I think it would be more logical to argue that PCT is more useful in severe CDI and more careful diagnosis in mild cases.

Reviewer 3 Report

Manuscript: Application of Procalcitonin, C-reactive Protein and White Blood Cell Count for the Rapid Diagnosis of Clostridium Difficile Infection in Patients with Inflammatory Bowel Disease

This statistical study propose the use of procalcitonin (PCT) and C-reactive Protein (CRP) as possible markers for rapid and no invasive diagnosis of Clostridium difficile in IBD patients. Authors found that CRP and PCT levels increase in IBD patients with Clostridium difficile infection as compare with controls (IBD only or IBD + viral/fungal infection). The White Blood Cell Count (WBC) was not a good diagnostic tool for Clostridium difficile infection, since this marker showed low sensitivity and low specificity.

The topic is relevant to find an easy diagnostic of Clostridium difficile in IBD patients. This statistical study analyzed a relative good number of patients for each group. An extensive introduction was included. However, this work will need minor modifications.  

Concerns and suggestions are listed below.

1.    Title must be modified, since it does not reflect what the authors found.  

2.    The result section is poorly written, you will need to improve it and be more explicit by describing the aim of each approach (receiver operating characteristic curve, sensitivity, specificity and positive likelihood ratio.

3.    Materials and Methods section must briefly describe the receiver operating characteristic curve, sensitivity%, specificity%, and positive likelihood ratio calculations.  

4.    Is the increment of CRP and PCT levels specific for Clostridium difficile infection? Can you include the samples of patient with other bacterial infection?     

5.    As the patient records were collected from The First Affiliated Hospital of Sun Yat-sen University.  Please discuss about the possibility to use this parameter different populations.

6.    Indicate asterisk (*) in the text accordingly with asterisk shown in table and figures.

Round 2

Reviewer 1 Report

Comments were addressed adequately, but some English language editing should be performed. Some word choices like enrolled suggests prospective collection when this is a retrospective study, so basic cleaning of the language should be performed.

Author Response

Comments were addressed adequately, but some English language editing should be performed. Some word choices like enrolled suggests prospective collection when this is a retrospective study, so basic cleaning of the language should be performed.

Response : We acknowledge your comments very much, which are valuable for improving the quality of our manuscript. According to your suggestion, we have checked the English expression of the whole manuscript and made necessary modifications.

We have tried our best to improve the manuscript. We appreciate for editors and reviewers’ warm work earnestly, and hope that the correction will meet with approval.

Once again, thank you very much for your comments and suggestions.

Yili Chen

Reviewer 2 Report

Unfortunately, I could not confirm that the peer review opinion has been properly corrected.

Anaerobe. 2016 Aug;40:95-9.  doi: 10.1016/j.anaerobe.2016.06.008. 

Round 3

Reviewer 2 Report

I have verified that it has been properly corrected.